# Modelling the economic burden of SARS-CoV-2 infection in health care workers in four countries

Huihui Wang[1,11], Wu Zeng [2,11] ✉, Kenneth Munge Kabubei[3], Jennifer J. K. Rasanathan[4], Jacob Kazungu[5], Sandile Ginindza[6], Sifiso Mtshali[7], Luis E. Salinas[8], Amanda McClelland[9], Marine Buissonniere[9], Christopher T. Lee[9], Jane Chuma[3], Jeremy Veillard[8], Thulani Matsebula[10] & Mickey Chopra[1]

Health care workers (HCWs) experienced greater risk of SARS-CoV-2 infection during the COVID-19 pandemic. This study applies a cost-of-illness (COI) approach to model the economic burden associated with SARS-CoV-2 infections among HCWs in five low- and middle-income sites (Kenya, Eswatini, Colombia, KwaZulu-Natal province, and Western Cape province of South Africa) during the first year of the pandemic. We find that not only did HCWs have a higher incidence of COVID-19 than the general population, but in all sites except Colombia, viral transmission from infected HCWs to close contacts resulted in substantial secondary SARS-CoV-2 infection and death. Disruption in health services as a result of HCW illness affected maternal and child deaths dramatically. Total economic losses attributable to SARS-CoV-2 infection among HCWs as a share of total health expenditure ranged from 1.51% in Colombia to 8.38% in Western Cape province, South Africa. This economic burden to society highlights the importance of adequate infection prevention and control measures to minimize the risk of SARS-CoV-2 infection in HCWs.

Though it is well-known and widely understood that COVID-19 is both a health crisis and economic crisis, the extent to which health care worker (HCW) SARS-CoV-2 infections pose a society-wide economic burden is less well understood. The burden of illness among HCWs includes costs of medical care, diminished personal earnings, and lost economic productivity over time[1]. When HCWs become infected in a pandemic, these costs are amplified by greater infectious spread, including in health care settings, within households of HCWs, and to the wider community[2].

In addition, however, high rates of SARS-CoV-2 infection among HCWs have the potential to generate a substantial, longer-term economic toll by disrupting the delivery of health services, such as

care for cancer patients and dialysis services, as well as maternal and child health services[3,4]. Consequences of the pandemic for maternal and child health care include but are not limited to fewer immunizations being given, fewer women receiving the full scope of antenatal care, and fewer babies being delivered in health care facilities[5,6]. Besides the service disruption due to SARS-CoV-2 infections among HCWs, the influx of COVID-19 patients and stringent control measures affect the delivery of essential services[3]. Economic costs associated with disruptions in health service delivery are borne by entire societies, especially in low- and middle-income countries (LMICs), where human resources for health are already in chronically short supply. Decades-long efforts to build human resources for

[1]World Bank, Washington, DC, USA. [2]Department of Global Health, Georgetown University, Washington, DC, USA. [3]World Bank Kenya Office, Nairobi, Kenya. [4]Independent Consultant, Geneva, Switzerland. [5]Health Economics Research Unit, KEMRI Welcome Trust Research Program, Nairobi, Kenya. [6]Pact, Mbabane, Eswatini. [7]Public Health Medicine Department, University of KwaZulu-Natal, Durban, South Africa. [8]World Bank Colombia Office, Bogota, Colombia. [9]Resolve to Save Lives, New York, NY, USA. [10]World Bank South Africa Office, Pretoria, South Africa. [11]These authors contributed equally: Huihui Wang, Wu Zeng. ✉e-mail: wz192@georgetown.edu

health are a testament that the health care workforce cannot be easily or rapidly generated[7].

Economic analyses have largely sought to quantify the overall cost of the COVID-19 pandemic[8], including the cost of lost productivity due to premature deaths from COVID-19[9], without specifically examining the economic burden attributed to SARS-CoV-2 infections in HCWs. Studies that have evaluated the economic cost of SARS-CoV-2 infections in HCWs tend to focus on HCW absenteeism due to COVID-19. One study in Iran, for example, calculated the economic cost of absenteeism in 1958 HCWs to be $1.3 million[10]. A cost-of-illness (COI) analysis in Greece considered only the cost of absenteeism (and presentism) along with the costs of direct medical care for infected HCWs[11].

To our knowledge, there is no comprehensive estimation of the society-wide economic burden in LMICs that captures the direct and indirect costs of SARS-CoV-2 infections among HCWs, the role of HCW infections in wider community transmission, and the economic toll of disrupted health services. Understanding the greater scale of economic costs may move national decision-makers and the global health community beyond panic-neglect cycles in pandemic response financing toward (re-)building a resilient health workforce and more sustainable, enduring pandemic preparedness, which includes adequate protection of HCWs.

This study, therefore, estimates the economic costs of SARS-CoV-2 infection in HCWs during the first year of the pandemic from the societal perspective in four LMICs. COVID-19 among HCWs results in enormous societal costs in these four countries. The economic costs of secondary infections and disruptions in essential health service delivery constitute a significant share of the total economic burden. Drawing on data from the first year of the COVID-19 pandemic, this study highlights the importance of adequate infection prevention and control measures to minimize the risk of SARS-CoV-2 infections among HCWs in the early stages of any pandemic.

## Results

### COVID-19 incidence among HCWs and COVID-19 cases and deaths by pathway

Figure 1 shows the COVID-19 incidence among HCWs in comparison with the general population in five sites. The COVID-19 incidence was higher in HCWs than in the general population in all study sites—almost 10 times higher in Kenya and 7–8 times higher in the two provinces of South Africa. Unlike other study sites, the COVID-19 incidence in HCWs was only slightly higher than in the general population (50.2 vs. 44.7 per 1000 population) in Colombia, which has the highest HCW density, comparatively. Supplementary Table 1 provides detailed information on COVID-19 cases and deaths among HCWs and the general population.

Table 1 shows the absolute number of SARS-CoV-2 infections and associated deaths accounted for in each pathway. Compared with the number of SARS-CoV-2 infections in HCWs, there were at least three times as many secondary infections in people who were in close contact with HCWs in all study sites except Colombia. Deaths due to secondary infections were greater than HCW deaths due to primary infection in every study site. There were more than 15 times as many deaths due to secondary infection as deaths from COVID-19 in HCWs in Western Cape province, South Africa. With respect to excess maternal and child mortality in the setting of health care workforce disruptions, there were an estimated 243 excess maternal deaths and 1499 excess deaths in children under five in Kenya, the highest toll across all study sites. Colombia had the second-highest number of excess maternal and child deaths, while maternal death rates in Eswatini and both provinces of South Africa were least affected by COVID-19-related disturbances to the health care workforce.

### Economic burden of SARS-CoV-2 infection in HCWs

Table 2 presents the economic costs associated with each pathway. Costs accrued along Pathway 1 ranged from US$2.00 million in

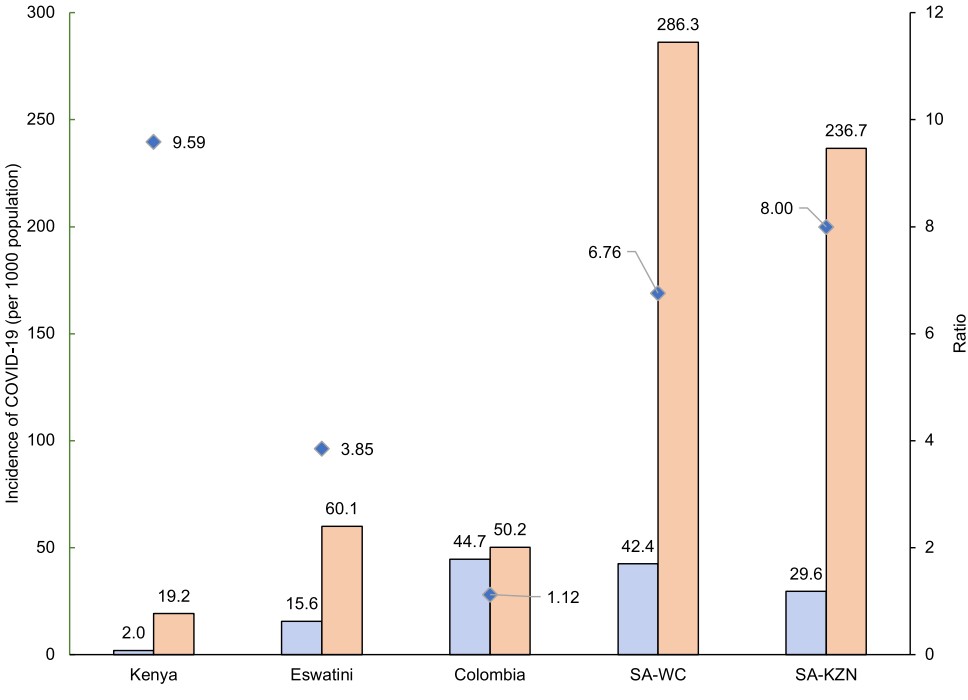

**Fig. 1 | SARS-CoV-2 infections among HCWs and across the general population in five sites.** It shows the incidence of COVID-19 in five sites among HCWs, in comparison with that among all populations, as well as the ratio of the two incidence rates. A ratio greater than one suggests a higher incidence among HCWs. SA South Africa; WC Western Cape; KZN KwaZulu-Natal.

**Table 1 | SARS-CoV-2 infections and/or deaths by pathway**

| | Kenya | Eswatini | Colombia | SA-WC | SA-KZN |
|---|---|---|---|---|---|
| *Pathway 1* | | | | | |
| Number of SARS-CoV-2 infections in HCWs | 3400 | 464 | 42,142 | 10,111 | 16,299 |
| Number of COVID-19 deaths in HCWs | 33 | 10 | 196 | 108 | 386 |
| *Pathway 2* | | | | | |
| The odds ratio of SARS-CoV-2 infection due to exposure to HCWs | 7.21 | 4.63 | 1.32 | 6.25 | 6.71 |
| Population attribution risk (PAR) | 9.42% | 14.20% | 1.94% | 13.90% | 20.30% |
| Secondary infections due to SARS-CoV-2 infection in HCWs | 9939 | 2607 | 43,786 | 41,162 | 69,331 |
| Secondary deaths due to SARS-CoV-2 infection in HCWs | 175 | 95 | 1177 | 1648 | 2141 |
| *Pathway 3* | | | | | |
| Maternal deaths | 243.0 | 6.0 | 29.0 | 4.0 | 8.0 |
| Deaths of children under 5 | 1499.0 | 34.0 | 235.0 | 70.0 | 206.0 |

The share of mild-moderate, severe, and critical COVID-19 cases was 81%, 14%, and 5%, respectively.

**Table 2 | Estimated economic burden by each pathway in 2020 US$ millions (percentage)**

| | Kenya | Eswatini | Colombia | SA-WC | SA-KZN |
|---|---|---|---|---|---|
| *Pathway 1* | | | | | |
| Medical costs | $2.79 (53.4%) | $0.79 (39.5%) | $65.64 (50.9%) | $19.99 (22.5%) | $32.22 (19.7%) |
| Non-medical costs | $0.04 (0.8%) | $0.01 (0.5%) | $1.44 (1.1%) | $0.20 (0.2%) | $0.32 (0.2%) |
| Indirect costs | $2.39 (45.8%) | $1.20 (60.0%) | $61.82 (48.0%) | $68.63 (77.3%) | $130.93 (80.1%) |
| Subtotal | $5.22 (100.0%) | $2.00 (100.0%) | $128.89 (100.0%) | $88.82 (100.0%) | $163.47 (100.0%) |
| *Pathway 2* | | | | | |
| Medical costs | $8.17 (54.6%) | $4.46 (44.9%) | $68.20 (28.1%) | $81.38 (34.4%) | $137.07 (39.8%) |
| Mon-medical costs | $0.10 (0.7%) | $0.06 (0.6%) | $1.49 (0.6%) | $0.81 (0.3%) | $1.36 (0.4%) |
| Indirect costs | $6.68 (44.7%) | $5.43 (54.6%) | $173.44 (71.3%) | $154.31 (65.2%) | $206.36 (59.9%) |
| Subtotal | $14.95 (100.0%) | $9.94 (100.0%) | $243.13 (100.0%) | $236.50 (100.0%) | $344.79 (100.0%) |
| *Pathway 3* | | | | | |
| Cost of maternal deaths | $12.00 (12.9%) | $0.55 (12.9%) | $5.37 (10.4%) | $0.62 (4.9%) | $1.19 (3.3%) |
| Cost of deaths in children under 5 | $81.03 (87.1%) | $3.70 (87.1%) | $46.47 (89.6%) | $11.98 (95.1%) | $35.19 (96.7%) |
| Subtotal | $93.03 (100.0%) | $4.25 (100.0%) | $51.84 (100.0%) | $12.60 (100.0%) | $36.38 (100.0%) |
| Total costs | $113.20 | $16.19 | $423.86 | $337.91 | $544.64 |

Eswatini to US$163.47 million in KwaZulu-Natal province, South Africa, alone, with a combined cost of US$252.29 million across both provinces in South Africa. Indirect costs comprised the largest share of costs associated with HCW infections and deaths in Eswatini as well as KwaZulu-Natal and Western Cape provinces in South Africa.

The total cost of secondary SARS-CoV-2 infections and related deaths accrued via pathway 2 in Eswatini was US$9.94 million, US$14.95 million in Kenya, and more than US$200 million for other study sites. Indirect costs due to lost productivity among contacts of HCWs with secondary infections—including people who either recovered or died of COVID-19—account for about half of the total costs of the secondary infections.

Maternal deaths were the least affected by HCW shortages in Eswatini and both provinces of South Africa compared to Kenya and Colombia. Kenya had the highest estimated economic loss of US$93.03 million through Pathway 3 due to excess maternal and child deaths. Colombia had the second-highest economic loss due to maternal and child deaths, amounting to US$51.84 million in economic losses.

The total economic burden of SARS-CoV-2 infections in HCWs along all pathways and the estimated cost of each HCW infection, including their 95% confidence intervals (CIs), are displayed in Table 3. Economic losses ranged from US$16.19 (95% CI: US$13.69-US$19.81) million in Eswatini to US$544.64 (95% CI: US$504.99-US$590.72)

million in KwaZulu-Natal province, South Africa, alone. These losses, as a share of total health expenditure, varied from 1.51% (95% CI: 1.39–1.68%) in Colombia to more than 8% of annual total health expenditure in the two provinces of South Africa. The associated economic burden in international dollars (I$) is presented in Supplementary Table 2.

There were different patterns of economic costs across the three pathways in the five study sites (Fig. 2). In Kenya, the costs of excess maternal and child deaths resulting from SARS-CoV-2 infections among HCWs (Pathway 3) represented the largest share of the total economic loss (82.2%), while the costs associated with secondary infections (Pathway 2) accounted for only 13.2% of the total economic cost. In Eswatini, Colombia, and both Western Cape and KwaZulu-Natal provinces of South Africa, where more than 5% of HCWs had SARS-CoV-2 infections, the costs along Pathway 2 accounted for the majority (57.4%-70.0%) of economic losses. In Colombia and both provinces of South Africa, the costs of primary HCW infections made up nearly a third of the total economic loss, while the economic costs associated with excess maternal and child deaths were relatively less substantial.

**Scenario and one-way sensitivity analysis**

Table 4 presents the total economic burden of SARS-CoV-2 infection among HCWs and the economic burden per HCW infection for both low- and high-impact scenarios. In the low-impact scenario, the total

**Table 3 | Total economic burden, economic cost per SARS-CoV-2 infection in HCWs, and their 95% CIs for the main analysis with 10,000 iterations**

|  | Total economic loss (US$ million) | Economic cost as a share of total health expenditure | Cost per SARS-CoV-2 infection (Total cost/ number of SARS-CoV-2 infections among HCWs) | GDP/ capita in 2020 US$ | Ratio of cost per SAR-CoV-2 infection among HCWs to GDP/capita |
|---|---|---|---|---|---|
| Kenya | $113.20 ($62.68–$190.34) | 2.03% (1.12–3.41%) | $33,619 ($18,616–$56,531) | $1870 | 17.98 (9.95–30.23) |
| Eswatini | $16.19 ($13.69–$19.81) | 5.01% (4.24–6.13%) | $35,659 ($30,158–$43,644) | $3941 | 9.05 (7.65–11.07) |
| Colombia | $423.86 ($390.25–$470.31) | 1.51% (1.39–1.68%) | $10,105 ($9,304–$11,212) | $6549 | 1.54 (1.42–1.71) |
| SA-WC | $337.91 ($302.48–$377.02) | 8.38% (7.50–9.35%) | $33,781 ($30,238–$37,691) | $5931 | 5.70 (5.10–6.35) |
| SA-KZN | $544.64 ($504.99–$590.72) | 8.21% (7.61–8.90%) | $34,226 ($31,735–$37,122) | $5931 | 5.77 (5.35–6.26) |

The numbers in the parathesis show the 95% CI of the estimate, respectively. *SA* South Africa; *WC* Western Cape; *KZN* KwaZulu-Natal. Total health expenditure in Western Cape and KwaZulu-Natal was estimated based on the share of the population in the two provinces.

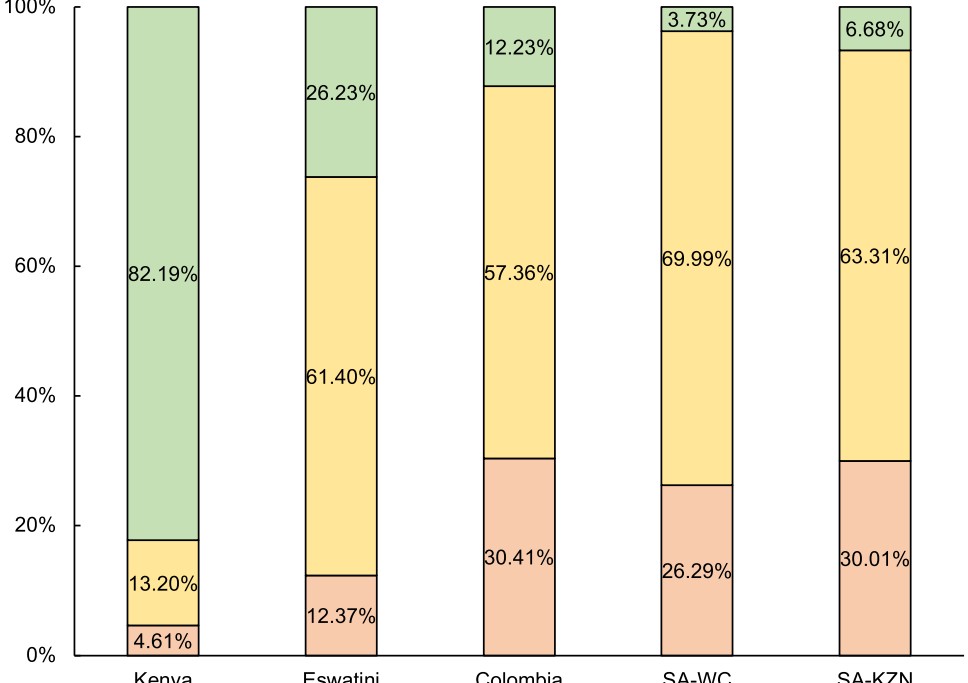

**Fig. 2 | Relative contribution of each pathway to the total economic cost of SARS-CoV-2 infection in HCWs in five study sites.** It illustrates the share of the total economic burden attributable to each pathway in five sites. SA South Africa; WC Western Cape; KZN KwaZulu-Natal.

economic losses ranged from US$11.58 million in Eswatini to US$471.96 million in KwaZulu-Natal province, South Africa, while these costs were US$23.21 million and US$633.22 million, respectively, in the high-impact scenario. As a share of total health expenditure, the economic losses ranged from 0.64% in Kenya to 7.11% in KwaZulu-Natal province, South Africa, in the low-impact scenario. In the high-impact scenario, economic losses as a share of total health expenditure were 1.82% in Colombia and up to 9.89% in Western Cape province, South Africa.

One-way sensitivity analysis shows the percentage change from the total economic costs we calculated when four parameters were adjusted (Supplementary Fig. 1). Kenya was most sensitive to variations in the parameters used to cost Pathway 3, while the South African provinces were most sensitive to variations in the proportion of inpatients considered close contacts of infected HCWs. The extent to which HCW productivity was impacted by SARS-CoV-2 infection substantially affected the estimated results in Eswatini and even more so in Kenya. The estimated cost along Pathway 3 in Kenya varied from 40.6% lower to 40.6% higher than the main analysis depending on this variable.

## Discussion

In this analysis, we have modeled the economic costs associated with SARS-CoV-2 infection in HCWs in the first year of the pandemic in three countries and two provinces of a fourth as they were incurred along three pathways. Unsurprisingly, SARS-CoV-2 infection in HCWs resulted in enormous societal costs, especially in settings where HCWs experienced disproportionately high rates of SARS-CoV-2 infection compared to the general population. Our findings corroborate other reports of higher SARS-CoV-2 infection rates in HCWs compared to the general population from all income settings and regions[12–14]. Sites in this study with substantial differences in the SARS-CoV-2 infection rate between HCWs and the general population bore the greatest financial toll as a percentage of total public health expenditure, with a societal 'price' per HCW infection that is several times higher than the per capita GDP. The economic costs associated with SARS-CoV-2 infection in HCWs are preventable if infectious risks to HCWs are mitigated upfront by working towards safer health facilities with full implementation of infection prevention and control (IPC) measures and water, sanitation, and hygiene (WASH) standards. There are robust IPC

**Table 4 | Total economic burden and economic cost per SARS-CoV-2 infection in HCWs for low- and high-impact scenarios**

| | Total economic loss (US$ million) | | Economic cost as a share of total health expenditure | | Cost per SARS-CoV-2 infection among HCWs (Total cost/ number of SARS-CoV-2 infections among HCWs) | | The ratio of cost per SARS-CoV-2 infection among HCWs to GDP/capita | |
|---|---|---|---|---|---|---|---|---|
| | Low-impact scenario | High-impact scenario | Low-impact scenario | High-impact scenario | Low-impact scenario | High-impact scenario | Low-impact scenario | High-impact scenario |
| Kenya | $35.83 | $246.12 | 0.64% | 4.41% | $10,641 | $73,098 | 5.69 | 39.09 |
| Eswatini | $11.58 | $23.21 | 3.59% | 7.19% | $25,509 | $51,121 | 6.47 | 12.97 |
| Colombia | $369.18 | $509.20 | 1.32% | 1.82% | $8,801 | $12,139 | 1.34 | 1.85 |
| SA-WC | $280.40 | $398.69 | 6.95% | 9.89% | $28,032 | $39,857 | 4.73 | 6.72 |
| SA-KZN | $471.96 | $633.22 | 7.11% | 9.54% | $29,659 | $39,793 | 5.00 | 6.71 |

SA South Africa; WC Western Cape; KZN KwaZulu-Natal. Total health expenditure in Western Cape and KwaZulu-Natal was estimated based on the share of the population in the two provinces.

standards and extensive normative occupational health guidance[15–17], reiterated by World Health Assembly resolution A74/A/CONF./6. Increased and coordinated investments in IPC training, supplies (including PPE), and monitoring—as well as adequate WASH facilities—are needed to support the full implementation of IPC guidance; implementation research may also help in this regard.

When HCWs are infected with SARS-CoV-2 (or other infectious pathogens), the health and economic impacts of those infections go far beyond the individual health and livelihoods of those HCWs. Although the cost of primary SARS-CoV-2 infection and related deaths in HCWs is not insignificant, the scale of economic losses mostly reflects onward infectious transmission from infected HCWs and disruptions in essential maternal and child health services as a result of HCW illness, isolation, or death. The extensive economic costs associated with disruptions in essential services estimated here are consistent with previous research documenting the sizable costs associated with HCW shortages in LMICs[18,19]. For example, the shortage of HCWs in LMICs due to the ongoing migration of physicians from LMICs to high-income countries is associated with an annual cost of US $15.86 billion as a result of mortality among children and pregnant women[18]. Additionally, a modeling study on the impact of the COVID-19 pandemic estimated that even small reductions in the availability of HCWs (for any reason), supplies as well as both demand for and access to health care would compromise a range of essential services and result in 24,400 additional maternal deaths and 417,000 additional child deaths per year globally[19]. These estimates not only demonstrate the need to closely monitor infection rates in HCWs during epidemics and pandemics but also highlight the importance of quantifying the economic costs of HCW infections and communicating the economic consequences of HCW infections effectively to the public.

Our modeled results show that maternal and child deaths due to compromised health service delivery contribute to a significant share of the total economic losses stemming from SARS-CoV-2 infections in HCWs. Countries with high rates of maternal and child mortality and inadequate human resources for health are likely to be vulnerable to even small changes in the health care workforce. Most maternal and child deaths are avoidable, and excess maternal and child deaths due to service disruptions are tragic reversals of earlier progress[20]. Of the countries and provinces included in this study, Eswatini and Kenya had the highest pre-pandemic under-five mortality rate (U5MR) and maternal mortality rate (MMR), and our cost estimates in Kenya and Eswatini were sensitive to the U5MR and MMR elasticities relative to HCW density as well as changes in productivity for HCWs remaining in post. In countries with high U5MR and MMR, there may be fewer other stopgaps in place to prevent unnecessary deaths when the health care workforce is (further) compromised. As maternal and child health services seem to be particularly sensitive to workforce disruptions during public health emergencies, dedicated measures to safeguard maternal and child health in countries with high baseline maternal and child mortality rates are critical. These might include task-shifting or bolstering child health with interventions that do not depend on HCW density[21]. Countries in this study implemented different measures to mitigate essential health service disruptions. Colombia developed a pandemic containment plan which included human resource retention strategies[22], and new HCWs were hired in Kenya's public sector[23]. The high costs incurred by excess maternal and child mortality highlight the importance of both protecting HCWs from infection and adopting other strategies to safeguard services that are sensitive to HCW density.

We have specifically considered the detrimental impact of SARS-CoV-2 infection among HCWs on the delivery of essential maternal and child health services, although many health services have been disrupted by the COVID-19 pandemic. Disruptions in maternal and child health services occurred in the first year of the COVID-19 pandemic for many reasons other than HCW infection, including, but not limited to,

decisions to suspend or reduce certain services or facilities; public health measures such as movement restrictions to 'flatten the curve' of SARS-CoV-2 infections; the surging volume of patients with suspected or confirmed COVID-19 in some settings or, elsewhere, sharp declines in patient attendance (for fear of infection or as a result of movement restrictions); supply chain interruptions; and HCW redeployment away from preventive to acute care services. All these factors contributed to a substantial increase in maternal and child deaths during the COVID-19 pandemic[19]. While HCW infections are not the only variable affecting service delivery, they can acutely worsen health outcomes by exacerbating already severe workforce shortages.

The COVID-19 pandemic has again spurred countries and the global development community to invest in and prioritize building resilient health systems[24]. Resilient health systems can adapt public health functions to mount a timely response to an infectious threat while protecting HCWs in order to preserve essential health service delivery. The economic cost of SARS-CoV-2 infections in HCWs as a percentage of total health expenditure was highest in the four study sites with the lowest HCW density. Efforts to maintain and adequately protect the health workforce during public health emergencies are, therefore, integral to strategies to strengthen health systems' resilience. All aspects of human resource production, deployment, and compensation should be oriented toward fortifying the health care workforce. Well-developed hazard compensation policies, for example, in Vietnam[25], demonstrate a recognition of the importance of maintaining an adequate health care workforce in times of crisis. Deployment of the health workforce during a crisis must be accompanied by comprehensive measures to support HCWs, including physical protection, psychological support, and child/family support. Many of these measures were implemented in various countries during the COVID-19 pandemic and require institutionalization[24]. Given the importance of HCWs for implementing public health emergency responses—and the ways in which lower HCW density can compromise essential health services—policies to attract, retain, and motivate qualified HCWs should be placed at the center of building more resilient health systems in LMICs.

The enormous economic cost associated with SARS-CoV-2 infections in HCWs as well as the moral imperative to protect HCWs during infectious outbreaks, demand accountability from governments, with guidance from WHO and adequate financing. While many countries adopted various strategies to protect HCWs during the pandemic and allocated resources to address the gaps and challenges[26–28], a holistic approach to protecting HCWs is needed. In addition to building and maintaining a core health care workforce, governments are responsible for mobilizing and allocating resources to protect health workers, ensuring proper use of these resources, and being accountable for the results.

Unlike many economic burdens of disease studies, which estimate the overall economic burden of a disease[29,30], we teased out the economic burden specifically attributable to SARS-CoV-2 infections in HCWs, which is one of the strengths of this study. Whereas this represents a fraction of the overall economic burden of the COVID-19 pandemic, the results of this modeling study are consistent with previous estimates of economic costs attributable to HCW infections, illness, and deaths in an infectious outbreak. For example, the economic burden due to HCW deaths and disruptions in health service delivery due to reduced HCW supply was estimated to be nearly double the costs of Ebola-related deaths in the 2014 Ebola virus disease outbreak[29].

The costs presented in this paper are likely to be conservative estimates of the cost to society of SARS-CoV-2 infections in HCWs for several reasons. First, it is likely that SARS-CoV-2 infections, related illnesses, and deaths were underreported. Under-reporting is common for infectious diseases generally, but challenges in accurately reporting SARS-CoV-2 infections and deaths are well recognized, including poor access to diagnostic tests in the first year of the pandemic, limited testing capacity, and inability to determine causality in the event of death[31]. Second, SARS-CoV-2 infections and related illnesses and deaths among HCWs are likely to have longer-term impacts on the health workforce pipeline that we did not attempt to capture. Third, we did not include SARS-CoV-2 infections in community health workers (CHWs) in this analysis because the five study sites lacked adequate data on both SARS-CoV-2 infection rates and the size of the CHW workforce. Economic losses would have been higher if CHWs were included in the analysis, especially in countries such as Kenya, Eswatini, and South Africa, where CHWs play an important role in health care delivery. Finally, we focused on three pathways through which HCW infections incur economic costs. While these pathways likely comprise the most important sources of economic losses, we were not able to quantify (1) the costs associated with worse health outcomes beyond excess maternal and child mortality, especially with respect to non-communicable diseases, which are prevalent; (2) nor does this analysis include the cost of training HCWs to replace those who are no longer working or alive as a result of SARS-CoV-2 infection; and (3) this study does not account for the costs associated with the mental health impacts of COVID-19 on HCWs or other long-term sequelae of infection such as long COVID[32].

Please note that all estimates presented in this paper pertain to the first year of the pandemic when there were substantial shortages of PPE, COVID-19 vaccination coverage among both the general population and HCWs was extremely low, and the capacity of health systems in some countries to respond to COVID-19 was quite limited. All these factors contributed to the potentially high economic costs presented in this paper. Since then, many circumstances have improved. With reduced virulence of the virus, much greater vaccine coverage, and enhanced treatment and testing capacities, the economic burden of HCW infections in subsequent years is likely to be substantially lower.

Several limitations of this study should be acknowledged. First, this study does not use an infectious disease transmission model to estimate the secondary infections from infected HCWs in each country. Instead, we estimated the odds ratio of infection for close contacts of HCWs based on an epidemiological study in a high-income country and used a log-linear regression to adjust for the difference in the relative risks faced by HCWs in each country. Further epidemiological research is likely to enable the global public health community to estimate the risk of SARS-CoV-2 transmission more precisely in different settings.

Second, data on SARS-CoV-2 infection and associated deaths in HCWs for the two sites in South Africa do not include data from the private sector. Data from the other study sites represent composite information from both the public and private sectors. If infections and deaths occurring in the private sector in Western Cape and KwaZulu-Natal provinces were included in this analysis, the economic costs would be even higher for Pathway 1. Limited evidence from Saudi Arabia suggests that SARS-CoV-2 infection rates may be similar between public and private facilities[33]. Assuming this is also true for the five sites in this study, the economic costs along Pathways 2 and 3 would not significantly vary. In this study, we do not extrapolate findings from the two provinces in South Africa to other provinces or to the whole country. However, it is important to understand the differences in the socio-economic (e.g., wages of HCWs) and health system situation (e.g., HCW density, MMR, and U5MR) between the two provinces and the whole country for any effort to use the parameters from the two provinces to estimate the economic burden for South Africa as a whole. While the model's estimates provide the total economic burden for each site, they don't account for variations within countries or individual sites. This is because the parameters used in the model represent site averages.

Third, some data were not available at the study sites. Thus, we had to make assumptions or draw from published literature from

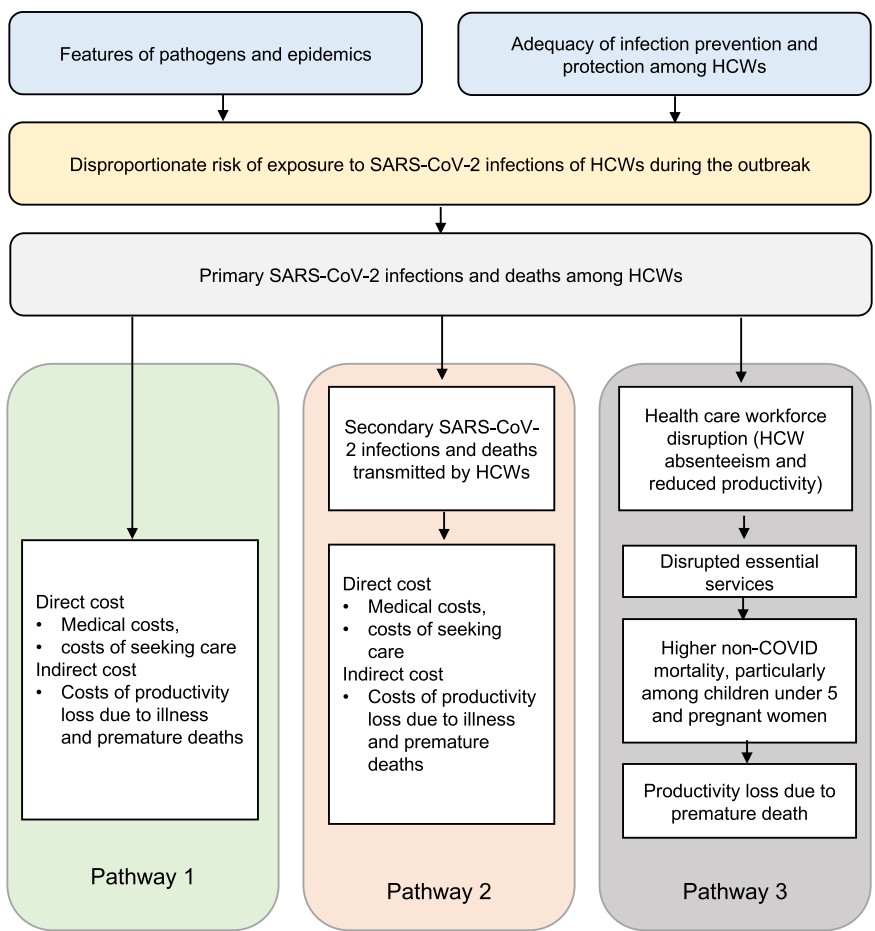

**Fig. 3 | Pathways from SARS-CoV-2 infection in HCWs to the economic cost.** It summarizes the three pathways through which the SARS-CoV-2 infection in HCWs incurred costs.

neighboring countries. Taking the cost per meal and cost of travel in Eswatini as an example, we used the analogous prices for South Africa as proxies when calculating direct non-medical costs. These inaccuracies are not likely to substantively change our results as we used the best available data to substitute for missing figures.

Fourth, some costs were not fully captured in the model. For example, estimating indirect costs due to loss of income would benefit from a more accurate rate of long-term absenteeism and the cost of HCW replacement. In this study, these costs are not included because data on the rates of long-term absenteeism are lacking. Additionally, the cost of presenteeism is not fully estimated. We assumed a 10% reduction in productivity among HCWs not infected (or not known to be infected) with the SARS-CoV-2 virus. However, the costs due to burnout and mental health impacts in HCWs are not included. Thus, the overall economic burden may be underestimated.

Fifth, in estimating potential service disruptions due to illness and deaths in HCWs, we did not account for steps that countries took to mitigate the impact of COVID-19 on their health workforce. Kenya hired more HCWs in the public sector[23], for example, and Colombia accelerated the validation of foreign qualifications and deployed medical students and graduates[22]. The impact of these actions on the estimated economic burden depends on the degree to which they actually boosted the health workforce in practice.

Finally, we could not precisely calculate the change in the productivity of HCWs who remained in their post in the first year of the COVID-19 pandemic. Productivity could vary by the number of colleagues absent, the number of deaths in HCW colleagues or family members, mental health status, the presence of post-covid symptoms,

and changes in patient demand. Instead, we assumed that all these factors reduced the overall productivity by an estimated 10% based on a prior study[19]. The sensitivity analysis shows that results from Kenya are sensitive to this assumption. Even under the low-impact scenario, which utilizes a 5% decrease in productivity, this still translates to US $10,641 lost per HCW infection, 5.7 times higher than per capita GDP.

## Methods

### Analysis framework

We developed a framework that lays out the major pathways through which SARS-CoV-2 infections in HCWs lead to population-wide morbidity and mortality based on a review of the literature regarding the health and economic impacts of SARS-CoV-2 infection in HCWs[10,12,34,35]. To quantify the economic burden of COVID-19 attributable to HCW infections, we followed the traditional COI approach to translate morbidity and mortality into the economic burden on the society[1] (Fig. 3).

The framework focuses on three pathways: (1) The first pathway accounts for the cost associated with primary SARS-CoV-2 infections and deaths among HCWs. HCWs in this study were defined as those who work in health institutions with professional health-related positions, including physicians, nurses, lab technicians, and health administrators. They did not include cleaners, drivers, or community health workers. Direct medical costs, direct non-medical costs (e.g., meals and transportation), and indirect costs associated with lost productivity due to illness and deaths were captured using the COI approach; (2) The second pathway accounts for the costs of secondary SARS-CoV-2 infections transmitted by infected HCWs and related

deaths. The same types of costs were estimated as in the first pathway; and (3) The third pathway encompasses the economic cost of excess deaths due to conditions other than COVID-19 as a result of health care workforce disruptions (e.g., absenteeism and reduced productivity due to stress, fatigue or missing essential team members). Although many services (e.g., cancer care, dialysis, and surgical treatment) have been affected by disruptions in the health workforce, this study focused on the impact of HCW infections on maternal and child mortality since the relationship between variations in the health workforce and maternal and child mortality is more established. Economic costs were estimated using lost human capital and lost years of economic productivity due to premature deaths of pregnant women and children under five years old. We estimated the economic costs incurred due to HCW infections during the first year of the COVID-19 pandemic. The study period was between March 1, 2020, and February 28, 2021, the first year of the pandemic in the five sites.

### Estimating infections and deaths under each pathway

*Pathway 1*: The number of known SARS-CoV-2 infections (symptomatic or not)—described here sometimes as COVID-19 cases—and related deaths in HCWs was collected through primary data collection. Each country research team liaised with national or provincial health authorities in charge of statistics on COVID-19 epidemiology and human resources for health to collect the data on SARS-CoV-2 infections and related deaths in the population and among HCWs in the study period, as well as the populations of various HCWs in the country/provinces. The numbers of SARS-CoV-2 infections and deaths among HCWs in each site are presented in the results section (Table 1).

*Pathway 2*: Close contacts of HCWs tend to have higher odds of infection with SARS-CoV-2 and of subsequent admission to hospitals[34]. The number of people with secondary infections due to transmission from an infected HCW was estimated by applying the concept of population-attributable risk (PAR), the proportion of SARS-CoV-2 infections in the general population that was attributable to close contact with an infected HCW in each study site. In this study, we defined close contacts as HCWs' household members and patients admitted to hospitals for inpatient care. We assumed 20% of inpatients were close contacts of HCWs in the main analysis, based on the contact intensity and the use of personal protective equipment (PPE) of HCWs when interacting with inpatients compared to that when interacting with their family members. The number of secondary infections was estimated to be the product of total SARS-CoV-2 infections and PAR, and the number of deaths due to secondary infections was estimated to be the product of the number of secondary infections and the case fatality rate in each site. PAR was calculated using the following formula:

$$PAR_i = \frac{E_i^*(OR_i - 1)}{E_i^*(OR_i - 1) + 1} \quad (1)$$

where *i* refers to site *i*, E is the share of the population considered to be close contacts of HCWs, and OR is the odds ratio of being diagnosed with COVID-19 due to the exposure to HCWs. Supplementary Methods present the detailed approach to derive E and OR.

*Pathway 3*: To estimate the additional child deaths resulting from health workforce disruptions as a result of SARS-CoV-2 infections and deaths in HCWs and productivity reduction among HCWs who remained on duty, we first converted the duration of HCW absence from work and the associated reduction in productivity into a decrease in HCW density. For HCWs who were not infected with SARS-CoV-2, we assumed a 10% reduction in their working productivity absent essential team members in the main analysis, using the same assumption as Roberton in the main analysis[19]. We estimated the resulting change in the U5MR based on its elasticity relative to HCW density[36] and, from

this, calculated additional deaths in children under five years old. We applied the same method to estimate excess maternal deaths, using the MMR and its elasticity relative to HCW density.

### Cost estimation

Once we estimated the number of infections and deaths in each pathway, we then applied the COI approach to estimate various costs. Direct cost estimation applied to all COVID-19 cases in Pathways 1 and 2 (whether people survived or died of the infection). The medical costs were estimated as the product of treatment cost per case and the number of cases by disease severity (e.g., mild-moderate, severe, and critical cases). The share of mild-moderate, severe, and critical cases was estimated to be 81%, 14%, and 5%, respectively[37], and we assumed that 80% of mild-moderate cases had home care while 20% of them received facility-based care. The non-medical costs of HCW infections included travel and meal costs while seeking or receiving facility-based care. As with the estimation of the direct medical costs, the direct non-medical costs were estimated as the sum of the product of the unit cost of travel and meals and the number of trips and meals across different severity levels of COVID-19 disease. For those who were infected with SARS-CoV-2 and survived, the indirect costs of their illness were estimated using the human capital approach for the period when they couldn't work due to COVID-19 disease (the product of their average daily wages and duration of absence from work). The average wages of HCWs were obtained from the literature or national agencies, while the average duration of absence (16.44 days) was obtained from the literature[10]. For those who died of COVID-19, we estimated the productivity losses associated with premature death. Gross Domestic Product (GDP) per capita in each country was used as a proxy for annual productivity loss[38,39], as the detailed age distribution and labor participation of those who died of COVID-19 were unknown. We estimated the number of productive years lost based on life expectancy at the age of death, according to the Global Health Observatory[40]. A discount rate of 3% was applied when estimating future productivity loss. All the costs were estimated in 2020 US dollars (US$).

### Site selection

Taking into consideration study feasibility and the importance of demonstrating the economic cost of HCW infection in a variety of settings, we included five study sites: Kenya, Eswatini, Colombia, and Western Cape and KwaZulu-Natal provinces of South Africa. The five sites were selected primarily based on data availability. In South Africa, we selected KwaZulu-Natal and Western Cape provinces because national aggregates were not available for South Africa, and these two provinces accounted for 40% of the national COVID-19 burden by February 2021. The study sites had diverse profiles with respect to their overall demographics, HCW density, SARS-CoV-2 infection rates, and COVID-19 mortality rates in both HCWs and the general population (Supplementary Table 3).

### Data sources

To estimate the costs of HCW infections according to the methodology described above, we used a mix of data sources: (a) primary data collection, in which the World Bank country offices obtained and verified data provided by national authorities on SARS-CoV-2 infection in HCWs and their outcomes (i.e., survival or death) in the first year of the pandemic, as well as average HCW income by profession to the extent possible; (b) World Development Indicators on country demographics, macroeconomic figures (e.g., GDP per capita) and health sector indicators (e.g., total health expenditure, total government health expenditure, hospital admission rates, under-five mortality rate, maternal mortality rate)[41]; (c) the Johns Hopkins University database on SARS-CoV-2 infections and deaths in the general population[42]; (d) peer-reviewed journal publications or gray

**Table 5 | Values of four parameters in each scenario**

| | Share of inpatients who are close contacts of HCWs (pathway 2) | Reduction in HCW productivity due to SARS-CoV-2 infection (pathway 3) | Elasticities of maternal and under-five mortality rates (pathway 3) |
|---|---|---|---|
| Low impact scenario | 10% | 5% | Mean + 1.96*SD |
| Moderate impact scenario (Main analysis) | 20% | 10% | Mean |
| High impact scenario | 30% | 15% | Mean − 1.96*SD |

*SD* standard deviation.

literature for other key parameters such as treatment cost per case, the composition of cases by severity level, length of stay in a medical facility, duration of absence from work; (e) the research team's own assumptions; and (f) parameter extrapolation from known sources, such as treatment costs for COVID-19 cases with various severity levels. Kenya was the only study site with an estimated treatment cost for home care for mild-moderate cases of COVID-19. Thus, the ratio of the treatment cost for home care to that for facility care among mild-moderate cases in Kenya was used to extrapolate the treatment cost of home care in other study sites. Similarly, if there were other unknown treatment costs in other sites, the corresponding cost ratios in Kenya were used to estimate the costs. Supplementary Tables 4-8 provide additional detail on all data sources and assumptions used for each study site.

### Scenario and sensitivity analysis
We created three scenarios based on different combinations of values in four parameters which are critical to final cost estimates and more likely subject to a range instead of fixed-point estimates: (1) the share of inpatients considered to be close contacts of HCWs; (2) the extent to which SARS-CoV-2 infection reduced health care workforce productivity; (3) the elasticity of MMR relative to HCW density; and (4) the elasticity of U5MR relative to HCW density. Table 5 shows the value of these 4 indicators in low, moderate, and high-impact scenarios. We used the moderate impact scenario for the main analysis and presented all findings under this scenario. We further conducted one-way sensitivity analyses of these four parameters by varying them from the low-impact value to the high-impact value individually and examined how they affect economic costs, reported as percentage changes from the main analysis.

To obtain 95% CIs for the moderate-impact scenario, we performed a stochastic sensitivity analysis on these four parameters and the treatment costs of COVID-19 cases with different severity levels. We used beta distribution for the share of inpatients considered close contacts of HCWs and the reduction in the health workforce productivity, with standard deviations assumed to be 20% of the means. Beta distribution was also applied to the elasticities of MMR and U5MR, with standard deviations obtained from the literature[36]. We used gamma distribution for treatment costs with standard deviations assumed to be 20% of the means[43], as their standard deviations were not available. The cost of meals and travel was not included in the stochastic sensitivity analysis, given their small share in the total economic burden. The 2.5th percentile and 97.5th percentile were obtained after we ran 10,000 iterations for these parameters, as the 95% CIs. All the simulations and analyses were performed using R (version 4.2.2).

### Reporting summary
Further information on research design is available in the Nature Portfolio Reporting Summary linked to this article.

## Data availability
The data on COVID-19 cases and deaths were reported in the results section, while the key parameters used for estimating the associated

economic burden were reported in Supplementary Tables 4–8. The data are also available at https://github.com/wuzengcn/HRH and https://doi.org/10.5281/zenodo.7856086.

## Code availability
The code used for this study is available at the following public GitHub repository: https://github.com/wuzengcn/HRH and https://doi.org/10.5281/zenodo.7856086.

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

## Acknowledgements
We are indebted to Mukesh Chawla from the World Bank for guiding the development of the costing framework.

## Author contributions
H.W., M.C., A.M., M.B., and C.T.L. conceptualized the study; H.W., W.Z., and M.C. developed the methodology; H.W., W.Z., K.M.K, J.J.K.R., J.K., S.G., S.M., L.E.S., J.C., J.V., and T.M. collected data; W.Z. and H.W. conducted data analysis; W.Z., J.J.K.R., H.W., A.M., M.B., and C.T.L. wrote the first draft of the paper, and all authors contributed to the critical review and revision of the paper.

## Competing interests
The authors declare no competing interests.
