## [Peer Review File · Nature Communications]

The economic burden of SARS-CoV-2 infection in health care workers in four countries: A modelling studyREVIEWER COMMENTS

Reviewer #1 (Remarks to the Author):

This paper entitled 'The economic burden of COVID-19 infections amongst health care workers in the first year of the pandemic in Kenya, Colombia, Eswatini, and South Africa' is noteworthy for the focus of the research on LIMICs (where standardised and comparable data can be difficult to source on key economic variables of interest), and in its efforts to estimate the indirect impact of Covid on Health Care Workers and those they came into contact with during this period. These secondary economic effects have not been captured in the literature to date in this geographical region, albeit the applied methodology has been widely used previously (and is therefore replicable and transparent). The work will therefore be of interest in the related epidemiological and public health fields providing an additional insight and perspective on the Covid burden, particularly in countries/regions that are often excluded from this type of analyses.

The work is cogent in nature, and the methodology sound, producing valid and reproducible results. The discussion and conclusion follows clearly from the results, although I do note a few points in my comments where greater discussion could be undertaken to shed more insight on the importance of the results in the literature. Useful sensitivity analysis is also provided based on key variables which are subject to uncertainty in their estimation.

Overall, I would recommend this paper for publication.

Introduction – No comments – relevant and focused introduction which provides key details and background on the objective of the research.

Data and Methodology

p.4, line 122 - These costs largely result from lost human capital and lost years of economic productivity due to premature deaths of pregnant women and children under 5.

I find the focus on disruption in health services as a result of HCW illness and how this affected maternal and child deaths as overly specialised. Why did the authors choose this particular focus, especially given the costing methodology employed (productivity costs tends to value lost work time)? The impact on, for example, cancer care screening and treatment services would be equally interesting and potentially monetarily large. Some rationale for this focus would be welcome.

p.4, line 132 – "from country data sources and is presented in the results section"

Please provide the Table reference here.

p.4, line 139 We assumed 20% of inpatients as close contacts of HCWs in the main analysis, based on the contact intensity of inpatients with HCWs compared to that of HCWs' family members with HCWs.

While this is noted as an assumption, is there any basis for it?

p. 5, line 174 For those who were infected with COVID-19 and survived, the indirect costs of their illness were estimated to be their lost income for the period when they couldn't work due to COVID-disease.

Were indirect costs valued according to the Human Capital Approach? If so this should be noted. It could also be noted – perhaps in the limitations section- that the friction cost approach may have derived a potentially more accurate estimate of the productivity costs where short terms absenteeism and long term absenteeism is involved, as we would assume worker replacement in

the health service case as alluded to in the discussion and therefore 'potential' productivity losses may be in excess of actual productivity losses.

p. 5, line 177 - Gross Domestic Product (GDP) per capita in each country was used as a proxy for annual productivity loss

This is in contrast to the approach to costing short term absences (employee income) and in contrast to much of the cost of illness literature (where national gross average or gender and age adjusted wage rates are used). Can you provide a rationale for this choice?

Results

Are the estimated costs in the Results section in dollar terms or in international dollar terms, the latter being more conducive to comparison across different countries for comparative purposes.

The use of percentages in Table 3 would better indicate the proportion of economic burden per category for the reader.

Presumably the Table 3 monetary values are in millions of dollars? This would need to be indicated similar to Table 4.

Discussion

p.8, line 322 - Substantial economic costs associated with disruptions in essential services disruption are consistent with previous research estimating the sizable costs associated with HCW shortages in LMICs.

Could the authors further expand upon this point? What are the differences between these study estimates and the Covid related estimates presented here?

p.10, line 377 - Many countries adopted various strategies to protect HCWs during the pandemic and specific resources were allocated to address the gaps and challenges.

This point would require references.

The discussion would benefit from a comparison of the costs estimated here and those associated with other diseases or disruption of HCW services due to other communicable illnesses. This comparison could be based on, for example, cost per infection, to provide greater perspective on the severity of the cost burden due to Covid.

p. 11, line 418 - Third, we did not include COVID-19 infections in community health workers in this analysis due to the absence of infection data from them

Can the authors provide an estimate for the size of the community health sector in the respective countries/regions?

The limitations section should also mention that presenteeism costs are also not captured by this paper's valuation.

Reviewer #2 (Remarks to the Author):

This is an important manuscript reporting on a study that estimated the economic burden associated with covid-19 amongst health care workers in a selection of 4 low- and middle-income countries (in the case of South Africa, two provinces only are included).

The paper is interesting and well-written, but while reading it I had a few major doubts:

1. Introduction: it highlights very much the role of HCW illness and death in the disruption and the delivery of healthcare services. While this is, of course, true, I felt that the role of other factors was not sufficiently highlighted. The disruption in healthcare services was also likely to be due to the large incremental burden on the healthcare sector directly due to covid-19 confirmed or suspected cases and to the fact that many services were suspended or reduced in order to avoid patients to go the health facilities and increase the burden on the HCW and also avoiding increasing transmission in health facility settings. Then, the study, focused on the primary covid-19 infections and deaths among HCWs and relative consequences; however, the sentence in the introduction saying "maternal and child health care delivery has been directly compromised globally by HCW illness and deaths..." sounds like an oversimplification of the situation.

2. Data and Methodology:

a. the selection of the sites is not sufficiently explained. Authors say that they purposively selected five study sites but they do not specify the criteria on which this choice was based;

b. I found it hard to understand at which level the analysis has been conducted, that is: I understand that the PAR is at the site level (?). However, E and OR are at the individual (HCW) level? In appendix 1 it is not clear what i is exactly: i is used for household size (which makes me think it indicates the single healthcare worker) but then it is also used for number of hospital admissions and for total population size...what is i , then?

c. The whole analysis is based in a number of strong assumptions and on a high level of uncertainty around the parameters used. The authors conduct a one way sensitivity analysis to resolve this. However, I think they should also conduct a probabilistic sensitivity analysis: the estimates they provide are highly uncertain and they should be presented with a confidence interval.

3. Discussion: authors mention the fact that Western Cape and KwaZulu-Natal provinces did not include data from the private sector. I think it would important also to highlight what is the socio-economic and health system situation of these two provinces with respect to the whole country. These are potentially, among the wealthiest provinces in South Africa which may have had an impact on the estimates.

4. Please be careful, the country is Colombia, not Columbia (see bottom page 6)

5. Another small note: COVID-19 is the disease, the infection is due to SARS-CoV-2, please check this over the manuscript and in the title as well

Reviewer #3 (Remarks to the Author):

Thank you for the opportunity to review this very interesting paper, which presents an estimate of the societal cost associated with COVID-19 infections among health care workers in the first year of the pandemic, in Kenya, Colombia, Eswatini, and South Africa. This is a very nice and well-done modelling analysis, which presents a thoughtful extrapolation of the potential costs associated with several aspects of HCW infection of COVID in the first year of the pandemic.

Comments:

Page 4, Pathway 1: Can you further describe how you estimated the number of COVID-19 infections in HCWs. What data was collected, from whom, and when?

Page 4, Pathway 2: The source you cite to justify that close contacts of HCWs have higher odds of infection with COVID-19 is about household members of HCWs, rather than inpatients in the hospital where HCWs work. I am not familiar with the literature on within-hospital transmission of COVID-19, however I would assume that transmission risk would be far lower in a hospital setting (where HCWs are required to wear PPE and take other preventative measures) than within the household, where exposure is prolonged and usually unprotected. Can you please add a citation to

justify your assumption of transmission of COVID-19 from HCWs to inpatients?

Page 5; Cost estimation: You mention that the indirect costs of illness for those who were infected with COVID-19 and survived were estimated as the lost income for the period when they couldn't work due to COVID-19 disease. Can you please describe how you estimated the duration of the period when they couldn't work due to COVID-19. Can you also clarify whether this was directly reported (actual) lost income, or whether this was estimated based on their daily salary and duration of illness.

Page 5; Cost estimation: Are health care workers given income protection (i.e. paid sick days) in any of the countries in the analysis? If so, the current approach to estimating income loss may be an over-estimate of productivity loss.

Page 5; Cost estimation: I know that most cost parameters are listed in the appendix tables, but I think it would be useful if you could describe assumptions for costs a bit further in the text. For example, how did you estimate the number of mild/moderate vs. severe cases from the total number of infections in each country? What proportion of mild/moderate cases were assumed to be treated at home vs. in the facility?

Page 5; Cost estimation: Are the parameter values reported in the appendix table mean or median estimates? Using a single parameter value may introduce a bias given that costs tend to be gamma-distributed. Perhaps consider estimating a range of costs/economic burden rather than an exact estimate, also so as not to miscommunicate the certainty of findings given that this is a broad-brush modelling analysis.

Page 6; Data sources: It looks like some key data (e.g. treatment cost for mild/moderate cases) was extrapolated from one country to another. Please could you include a few sentences describing your methods for extrapolation.

Table 2: It might be useful to see the total number of COVID-19 infections broken down into mild/moderate vs. severe cases, if this data are available.

Table 3: Please add a label to note the values are in millions, and clarify currency year.

Table 4: I find it slightly misleading to present the low impact scenario and high impact scenario in parentheses in this table – as these are results from your scenario analyses rather than a representation of confidence intervals or certainty from your estimates. I would suggest to present your scenario analyses separately from your base case analysis, but also to provide a range of cost estimates e.g. using ranges in cost inputs to your model.

Discussion: The analysis is very broad-brush, using a combination of primary and secondary data, supplemented with assumptions where no data exists. While the breadth of the analysis is not an issue in itself, it would be good to make this very clear in the presentation and discussion of results to avoid misinterpretation by policymakers and other stakeholders. I'd suggest revising some wording in the discussion to make clear that these results are not a presentation of scientific fact, but rather a modelling analysis designed to identify the potential scale of the issue and important considerations in responding to the pandemic.

Discussion: I would suggest to make clear throughout the paper that this analysis was done using data from the first year of the pandemic. Many things have changed very rapidly since then (including infection control measures, infectiousness of COVID-19 strains, vaccine availability etc.) and the timeframe of the analysis is very important for interpretation of results. It would be useful to add a reflection as to what has changed in the last couple of years, and what the results of the analysis might mean in terms of policy today.

Response to reviewers' comments

We appreciate the reviewers' insightful comments. Below we explain, point by point, how we dealt with the comments from the reviewers. To make it easier for the reviewers, we have left the text of the reviewers and added our response in blue right below each of the comments.

REVIEWER COMMENTS

Reviewer #1 (Remarks to the Author):

This paper entitled 'The economic burden of COVID-19 infections amongst health care workers in the first year of the pandemic in Kenya, Colombia, Eswatini, and South Africa' is noteworthy for the focus of the research on LIMICs (where standardised and comparable data can be difficult to source on key economic variables of interest), and in its efforts to estimate the indirect impact of Covid on Health Care Workers and those they came into contact with during this period. These secondary economic effects have not been captured in the literature to date in this geographical region, albeit the applied methodology has been widely used previously (and is therefore replicable and transparent). The work will therefore be of interest in the related epidemiological and public health fields providing an additional insight and perspective on the Covid burden, particularly in countries/regions that are often excluded from this type of analyses.

The work is cogent in nature, and the methodology sound, producing valid and reproducible results. The discussion and conclusion follows clearly from the results, although I do note a few points in my comments where greater discussion could be undertaken to shed more insight on the importance of the results in the literature. Useful sensitivity analysis is also provided based on key variables which are subject to uncertainty in their estimation.

Overall, I would recommend this paper for publication.

Introduction – No comments – relevant and focused introduction which provides key details and background on the objective of the research.

Data and Methodology

p.4, line 122 - These costs largely result from lost human capital and lost years of economic productivity due to premature deaths of pregnant women and children under 5.

I find the focus on disruption in health services as a result of HCW illness and how this affected maternal and child deaths as overly specialized. Why did the authors choose this particular focus, especially given the costing methodology employed (productivity costs tends to value lost work time)? The impact on, for example, cancer care screening and treatment services would be equally interesting and potentially monetarily large. Some rationale for this focus would be welcome.

Response: Thanks for the comments. We agree with the reviewer's observation that there are many other conditions affected by COVID-19 among HCWs. The reason for focusing on MCH services was that there is an established relationship between maternal and child mortality rates and HCWs density, which was measured by the HWC density elasticity of mortalities.

To address the reviewer's concern, we have revised the sentences and added more explanation: *“Although many services (e.g., cancer care, dialysis, and surgical treatment) have been affected by disruptions in the health workforce, this study focused on the impact of HCW infections on maternal and child mortality since the relationship between variations in the health workforce and maternal and child mortality is more established.”*

p.4, line 132 – “from country data sources and is presented in the results section”

Please provide the Table reference here.

Response: As suggested, a table reference is added.

p.4, line 139 We assumed 20% of inpatients as close contacts of HCWs in the main analysis, based on the contact intensity of inpatients with HCWs compared to that of HCWs' family members with HCWs.

While this is noted as an assumption, is there any basis for it?

Response: Thanks. There are two major factors that we considered: one is the difference in contact time between family members and inpatients, and the other is the difference in using personal protection equipment (PPE) at home and in health facilities. We assumed that family members' contact time was 2 times more than inpatients' interaction with HCWs. Unfortunately, we cannot find any literature on this. We basically assumed that HCWs spent 8 hours working in health facilities and 16 hours at home. HCWs were less like to wear PPE when interacting with family members than with inpatients. We assumed 30% of PPE shortage¹ in health facilities, and a protection rate of 80% for PPE². When taking all these factors into consideration, the ratio of the risk of HCW being infected when contacting inpatients to that when contacting family members is estimated as $(1*0.3*1+1*0.7*(1-0.8))/(2*1)$, which is 22%. We made a conservative estimate and round it to 20%. Given the uncertainty, we conducted a one-way sensitivity analysis of this parameter.

We have revised the sentence to *“We assumed 20% of inpatients as close contacts of HCWs in the main analysis, based on the contact intensity and the use of personal protective equipment (PPE) of HCWs when interacting with inpatients compared to that when interacting with their family members”*.

p. 5, line 174 For those who were infected with COVID-19 and survived, the indirect costs of their illness were estimated to be their lost income for the period when they couldn't work due to

¹ Tabah et al. 2020. *Personal protective equipment and intensive care unit healthcare worker safety in the COVID-19 era (PPE-SAFE): An international survey. Journal of Critical Care 59:70-75*

² Criswold, et al. 2021. *Personal protective equipment for reducing the risk of COVID-19 infection among health care workers involved in emergency trauma surgery during the pandemic: An umbrella review. J Trauma Acute Care Surg 90 (4):72-80*

COVID-disease.

Were indirect costs valued according to the Human Capital Approach? If so this should be noted. It could also be noted – perhaps in the limitations section- that the friction cost approach may have derived a potentially more accurate estimate of the productivity costs where short term absenteeism and long term absenteeism is involved, as we would assume worker replacement in the health service case as alluded to in the discussion and therefore ‘potential’ productivity losses may be in excess of actual productivity losses.

Response: Thanks for the suggestions. Yes, it is also the human capital approach. Following your suggestion, we have added “*using the human capital approach*” to the sentence.

As suggested, we added the following sentences in the limitation section: “*Fourth, some costs were not fully captured in the model. For example, estimating indirect costs due to loss of income would benefit from a more accurate rate of long-term absenteeism and the cost of HCW replacement. In this study, these costs are not included because data on the rates of long-term absenteeism are lacking.*”

p. 5, line 177 - Gross Domestic Product (GDP) per capita in each country was used as a proxy for annual productivity loss

This is in contrast to the approach to costing short term absences (employee income) and in contrast to much of the cost of illness literature (where national gross average or gender and age-adjusted wage rates are used). Can you provide a rationale for this choice?

Response: Thanks for pointing it out. For the short-term absence specific for health care workers, who are in the labor force, we collected information on the average wage for each type of HCWs. Using wages is more accurate in estimating the economic loss due to the short-term absence.

For premature deaths, wage³ and GDP/capita^{4,5} approaches are both popular to estimate the associated indirect cost. When estimating the economic loss due to the deaths among HCWs, pregnant women and children, and the general population through the three pathways, the affected population is mixed. We were not able to obtain detailed information on their age distribution and labor participation. Thus, we used GDP/capita to estimate the indirect costs to avoid adding more uncertainties to the model.

We added two references to the sentence and added the phrase of “*as the detailed age distribution and labor participation of those who died of COVID-19 were unknown*” to justify the use of GDP/capita for the estimation.

Results

³ Reuter A. 2022. Global economic burden of unmet surgical need for appendicitis. Br J Surg 109(10):995-1003

⁴ Ding D, et al. 2016. The economic burden of physical inactivity: a global analysis of major non-communicable diseases. Lancet 388: 1311-24

⁵ Bommer C, et al. 2017. The global economic burden of diabetes in adults aged 20-79 years: a cost-of-illness study. Lancet Diabetes Endocrinol 5(6): 423-430

Are the estimated costs in the Results section in dollar terms or in international dollar terms, the latter being more conducive to comparison across different countries for comparative purposes.

Response: Currently, the economic burden was estimated in US dollars in 2020.

We added, in the method section, the phrase: “*All the costs were estimated in 2020 US dollars (US\$).*” To avoid overcrowding the main text and the associated table, we provided the results in international dollars in appendix 5.

The use of percentages in Table 3 would better indicate the proportion of economic burden per category for the reader.

Response: As suggested, we added the proportion to the table.

Presumably the Table 3 monetary values are in millions of dollars? This would need to be indicated similar to Table 4.

Response: Yes, it is in millions of dollars. We have added it to the title of Table 4.

Discussion

p.8, line 322 - Substantial economic costs associated with disruptions in essential services disruption are consistent with previous research estimating the sizable costs associated with HCW shortages in LMICs.

Could the authors further expand upon this point? What are the differences between these study estimates and the Covid related estimates presented here?

Response: Following the reviewer’s suggestion, we have added a further explanation on this point. Please note, in our study, we tried to limit the economic burden of disruption of health services due to presenteeism and absenteeism of HCWs only. The cost associated with the service disruptions due to other reasons such as lockdown was not included.

We have added the following sentences to the paragraph: “*For example, the shortage of HCWs in LMICs due to the ongoing migration of physicians from LMICs to high-income countries is associated with an annual cost of \$15.86 billion as a result of mortality among children and pregnant women.¹⁸ Additionally, a modelling study on the impact of the COVID-19 pandemic estimated that even small reductions in the availability of HCWs (for any reason), supplies as well as both demand for and access to health care would compromise a range of essential services and result in 24,400 additional maternal deaths and 417,000 additional child deaths per year globally.¹⁹”*

p.10, line 377 - Many countries adopted various strategies to protect HCWs during the pandemic and specific resources were allocated to address the gaps and challenges.

This point would require references.

Response: Thanks for the suggestion, we have added a few references.

The discussion would benefit from a comparison of the costs estimated here and those associated with other diseases or disruption of HCW services due to other communicable illnesses. This comparison could be based on, for example, cost per infection, to provide greater perspective on the severity of the cost burden due to Covid.

Response: Thanks for the suggestion. We have further conducted a search of articles. We found that a majority of economic burden studies focus on the overall burden of disease. Few estimated the cost per infection as we did in this paper. However, the cost due to the impact on HCWs shared a significant part of the total economic burden. We have added the following sentence in the discussion section:

“Unlike many economic burden of disease studies which estimate the overall economic burden of a disease,^{29,30} we teased out the economic burden specifically attributable to SARS-CoV-2 infections in HCWs, which is one of the strengths of this study. Whereas this represents a fraction of the overall economic burden the COVID-19 pandemic, the results of this modelling study are consistent with previous estimates of economic costs attributable to HCW infections, illness, and deaths in an infectious outbreak. For example, the economic burden due to HCW deaths and disruptions in health service delivery due to reduced HCW supply was estimated to be nearly double the costs of Ebola-related deaths in the 2014 Ebola virus disease outbreak.²⁹”

p. 11, line 418 - Third, we did not include COVID-19 infections in community health workers in this analysis due to the absence of infection data from them

Can the authors provide an estimate for the size of the community health sector in the respective countries/regions?

Response: Thanks for the suggestion. We have further requested the local team to obtain the size of community health workers (CHWs) in the five sites. Unfortunately, in all five sites, we could not obtain an official number of healthcare workers. Nor do we have information on the infection and deaths among them, and thus it makes it difficult to have an accurate estimation of the economic burden due to CHWs. Our qualitative assessment is that the economic burden would be even higher if CHWs were included through the same three pathways.

We have added the following sentence: *“Economic losses would have been higher if CHWs were included in the analysis, especially in countries such as Kenya, Eswatini, and South Africa where CHWs play an important role in health care delivery.”*

The limitations section should also mention that presenteeism costs are also not captured by this paper’s valuation.

Response: Presenteeism is partially captured in this study. We assumed that those present in the work will have productivity compromised by 10%, which was assumed based on the article published by Robertson et al. To make it more explicit, we have added *“Additionally, the cost of*

presenteeism is not fully estimated. We assumed a 10% reduction in productivity among HCWs not infected (or not known to be infected) with SARS-CoV-2 virus. However, the costs due to burnout and mental health impacts in HCWs are not included. Thus, the overall economic burden may be an underestimate.”

Reviewer #2 (Remarks to the Author):

This is an important manuscript reporting on a study that estimated the economic burden associated with covid-19 amongst health care workers in a selection of 4 low- and middle-income countries (in the case of South Africa, two provinces only are included).

The paper is interesting and well-written, but while reading it I had a few major doubts:

1. Introduction: it highlights very much the role of HCW illness and death in the disruption and the delivery of healthcare services. While this is, of course, true, I felt that the role of other factors was not sufficiently highlighted. The disruption in healthcare services was also likely to be due to the large incremental burden on the healthcare sector directly due to covid-19 confirmed or suspected cases and to the fact that many services were suspended or reduced in order to avoid patients to go the health facilities and increase the burden on the HCW and also avoiding increasing transmission in health facility settings. Then, the study, focused on the primary covid-19 infections and deaths among HCWs and relative consequences; however, the sentence in the introduction saying “maternal and child health care delivery has been directly compromised globally by HCW illness and deaths...” sounds like an oversimplification of the situation.

Response: Thanks for the comment. We agree with the reviewer. We added two more examples on the service disruptions, and slightly downplay the service disruption on maternal and child health services.

The revised paragraph now reads as *“In addition, however, high rates of SARS-CoV-2 infection among HCWs have the potential to generate a substantial, longer-term economic toll by disrupting delivery of health services, such as care for cancer patients and dialysis services, as well as maternal and child health services.^{3,4} Consequences of the pandemic for maternal and child health care include, but are not limited to, fewer immunizations being given, fewer women receiving the full scope of antenatal care, and fewer babies being delivered in health care facilities.^{5,6} Besides the service disruption due to SARS-CoV-2 infections among HCWs, the influx of COVID-19 patients and stringent control measures affect the delivery of essential services.³”*

We also addressed this comment by adding the following text to the discussion:

“We have specifically considered the detrimental impact of SARS-CoV-2 infection among HCWs on the delivery of essential maternal and child health services, although many health services have been disrupted by the COVID-19 pandemic. Disruptions in maternal and child health services occurred in the first year of the COVID-19 pandemic for many reasons other than HCW

infection including, but not limited to: decisions to suspend or reduce certain services or facilities; public health measures such as movement restrictions to ‘flatten the curve’ of SARS-CoV-2 infections; surging volume of patients with suspected or confirmed COVID-19 in some settings or, elsewhere, sharp declines in patient attendance (for fear of infection or as a result of movement restrictions); supply chain interruptions; and HCW redeployment away from preventive to acute care services. All these factors contributed to a substantial increase in maternal and child deaths during the COVID-19 pandemic.¹⁹ While HCW infections are not the only variable affecting service delivery, they can acutely worsen health outcomes by exacerbating already severe workforce shortages.”

2. Data and Methodology:

a. the selection of the sites is not sufficiently explained. Authors say that they purposively selected five study sites but they do not specify the criteria on which this choice was based;

Response: The selection of the study site is primarily based on the availability of the data on the size and epidemiological information on COVID-19 among them, and the pandemic profile. The study lead at the World Bank sent an email to many country world bank teams to solicit their interest in participating in the study and reviewed the initial data collected from the field. Kenya, Colombia, South Africa, and ESwatini are the four countries that expressed interest and had reasonably complete information for the modelling.

We have added the following sentences to the paragraph: *“The five sites were selected primarily based on the data availability”* We also revised the sentence *“we purposefully selected five study sites”* to *“we included five study sites.”*

b. I found it hard to understand at which level the analysis has been conducted, that is: I understand that the PAR is at the site level (?). However, E and OR are at the individual (HCW) level? In appendix 1 it is not clear what **i** is exactly: i is used for household size (which makes me think it indicates the single healthcare worker) but then it is also used for number of hospital admissions and for total population size...what is i, then?

Response: All the analyses were at the site level. *i* refers to the site *i*. For example, household size_{*i*} is the average household size in site *i*. To avoid confusion, we have added “*i*” to the formula for PAR and “*i* refers to site *i*” in the main text.

In the appendix, we explained the meaning of E_{*i*}: *“the share of the population considered to be close contacts of HCWs for site *i*”*, and also added the following sentence to explain the formula for OR: *“where *i* refers to site *i*, and *r* refers to the reference country --- Scotland.”*

c. The whole analysis is based in a number of strong assumptions and on a high level of uncertainty around the parameters used. The authors conduct a one-way sensitivity analysis to resolve this. However, I think they should also conduct a probabilistic sensitivity analysis: the estimates they provide are highly uncertain and they should be presented with a confidence interval.

Response: Thanks for the suggestion. As suggested, we have added probability sensitivity results in the text, around the four major parameters and treatment costs. The added text in the methodology section reads as *“To obtain 95% confidence intervals (CIs) for the moderate-impact scenario, we performed a stochastic sensitivity analysis on these four parameters and the treatment costs of COVID-19 cases with different severity levels. We used beta distribution for the share of inpatients considered close contacts of HCWs and the reduction in the health workforce productivity, with a standard deviation assumed to be 20% of the mean. Beta distribution was also applied to the elasticities of MMR and U5MR, with standard deviations obtained from the literature.³⁶ We used gamma distribution for treatment costs with standard deviations assumed to be 20% of the means,⁴³ as their standard deviations were not available. The cost of meals and travel was not included in the stochastic sensitivity analysis given their small share in the total economic burden. The 2.5th percentile and 97.5th percentile were obtained after we ran 10,000 iterations for these parameters, as the 95% CIs.”*

We also revised the result section accordingly.

3. Discussion: authors mention the fact that Western Cape and KwaZulu-Natal provinces did not include data from the private sector. I think it would be important also to highlight what is the socio-economic and health system situation of these two provinces with respect to the whole country. These are potentially, among the wealthiest provinces in South Africa which may have had an impact on the estimates.

Response: Thanks for the comment. We believe that the socio-economic and health system situation would be very important if the results from the two provinces were used to extrapolate the estimate for the whole country. However, in this study, we did not extrapolate the estimates from the two provinces to South Africa as a whole.

To make it clearer, we added the following to the discussion. *“In this study, we do not extrapolate findings from the two provinces in South Africa to other provinces or to the whole country. However, it is important to understand the differences in the socio-economic (e.g., wages of HCWs) and health system situation (e.g., HCW density, MMR, and U5MR) between the two provinces and the whole country for any effort to use the parameters from the two provinces to estimate the economic burden for South Africa as a whole.”*

4. Please be careful, the country is Colombia, not Columbia (see bottom page 6)

Response: Thanks for catching the typo. We have corrected it as suggested.

5. Another small note: COVID-19 is the disease, the infection is due to SARS-CoV-2, please check this over the manuscript and in the title as well

Response: Thanks. We have corrected it as suggested throughout the manuscript.

Reviewer #3 (Remarks to the Author):

Thank you for the opportunity to review this very interesting paper, which presents an estimate

of the societal cost associated with SARS-CoV-2 infections among health care workers in the first year of the pandemic, in Kenya, Colombia, Eswatini, and South Africa. This is a very nice and well-done modelling analysis, which presents a thoughtful extrapolation of the potential costs associated with several aspects of HCW infection of COVID in the first year of the pandemic.

Comments:

Page 4, Pathway 1: Can you further describe how you estimated the number of COVID-19 infections in HCWs. What data was collected, from whom, and when?

Response: Thanks, we have revised the paragraph, which now reads as “*The number of known SARS-CoV-2 infections (symptomatic or not) – described here sometimes as COVID-19 cases – and related deaths in HCWs was collected through primary data collection. Each country research team liaised with national or provincial health authorities in charge of statistics on COVID-19 epidemiology and human resources for health, to collect the data on SARS-CoV-2 infections and related deaths in the population and among HCWs in the study period, as well as the populations of various HCWs in the country/provinces. The numbers of SARS-CoV-2 infections and deaths among HCWs in each site are presented in the results section (Table 2).*”

Page 4, Pathway 2: The source you cite to justify that close contacts of HCWs have higher odds of infection with COVID-19 is about household members of HCWs, rather than inpatients in the hospital where HCWs work. I am not familiar with the literature on within-hospital transmission of COVID-19, however I would assume that transmission risk would be far lower in a hospital setting (where HCWs are required to wear PPE and take other preventative measures) than within the household, where exposure is prolonged and usually unprotected. Can you please add a citation to justify your assumption of transmission of COVID-19 from HCWs to inpatients?

Response: Thanks. The effect of using PPE among HCWs when interacting with patients is partially captured by the assumption that 20% of inpatients are considered close contacts of HCWs. The justification was raised by reviewer 1.

To ease your review, we repeat our response to reviewer 1 regarding the justification of using 20% of inpatients as close contacts of HCWs below.

*There are two major factors that we considered: one is the difference in contact time between family members and inpatients, and the other is the difference in using personal protection equipment (PPE) at home and in health facilities. We assumed that family members' contact time was 2 times more than inpatients' interaction with HCWs. Unfortunately, we cannot find any literature on this. We basically assumed that HCWs spent 8 hours working in health facilities and 16 hours at home. HCWs were less like to wear PPE when interacting with family members than with inpatients. We assumed 30% of PPE shortage⁶ in health facilities, and a protection rate of 80% for PPE⁷. When taking these two factors into consideration, the ratio of the risk of being infected in inpatients to that in family members is estimated as $(1*0.3*1+1*0.7*(1-0.8))$*

⁶ Tabah et al. 2020. *Personal protective equipment and intensive care unit healthcare worker safety in the COVID-19 era (PPE-SAFE): An international survey. Journal of Critical Care 59:70-75*

⁷ Criswold, et al. 2021. *Personal protective equipment for reducing the risk of COVID-19 infection among health care workers involved in emergency trauma surgery during the pandemic: An umbrella review. J Trauma Acute Care Surg 90 (4):72-80*

*(2*1), which is 22%. We made a conservative estimate and round it to 20%. Given the uncertainty, we conducted a one-way sensitivity analysis on this parameter.*

We have revised the sentence to “We assumed 20% of inpatients as close contacts of HCWs in the main analysis, based on the contact intensity and the use of personal protective equipment (PPE) of HCWs when interacting with inpatients compared to when interacting with their family members”.

We hope this addresses your concern.

Page 5; Cost estimation: You mention that the indirect costs of illness for those who were infected with COVID-19 and survived were estimated as the lost income for the period when they couldn't work due to COVID-19 disease. Can you please describe how you estimated the duration of the period when they couldn't work due to COVID-19. Can you also clarify whether this was directly reported (actual) lost income, or whether this was estimated based on their daily salary and duration of illness.

Response: Thanks for the comment. The duration of being absent from work was estimated to be 16.44 days from the literature. The cost was estimated as the product of the average daily salary and the duration of the illness.

We have added the following sentences: *“(the product of their average daily wages and duration of absence from work). The average wages of HCWs were obtained from the literature or national agencies, while the average duration of absence (16.44 days) was obtained from the literature.¹⁰”* We also added a reference.

Page 5; Cost estimation: Are health care workers given income protection (i.e. paid sick days) in any of the countries in the analysis? If so, the current approach to estimating income loss may be an over-estimate of productivity loss.

Response: Thanks for the comment. From the societal point of view, even if the sick HCWs are paid because of the income protection, being sick that leads to productivity loss is a loss to society. Thus, the lost income should be estimated and included in the economic loss, no matter whether HCWs are paid or not.

Page 5; Cost estimation: I know that most cost parameters are listed in the appendix tables, but I think it would be useful if you could describe assumptions for costs a bit further in the text. For example, how did you estimate the number of mild/moderate vs. severe cases from the total number of infections in each country? What proportion of mild/moderate cases were assumed to be treated at home vs. in the facility?

Response: Thanks for the suggestion. We have added more parameters in the main text. We added the following sentences to the cost estimation paragraph: *“The share of mild-moderate, severe, and critical cases was estimated to be 81%, 14%, and 5%, respectively,³⁷ and we assumed that 80% of mild-moderate cases had home care while 20% of them received facility-based care”*

Page 5; Cost estimation: Are the parameter values reported in the appendix table mean or median estimates? Using a single parameter value may introduce a bias given that costs tend to be gamma-distributed. Perhaps consider estimating a range of costs/economic burden rather than an exact estimate, also so as not to miscommunicate the certainty of findings given that this is a broad-brush modelling analysis.

Response: Thanks for the information. The costs from the literature are generally means. Following your suggestions and the suggestion from reviewer 2, we have conducted a probabilistic sensitivity analysis. We added some text to the methodology and result section.

The added text in the methodology section is *“To obtain 95% confidence intervals (CIs) for the moderate-impact scenario, we performed a stochastic sensitivity analysis on these four parameters and the treatment costs of COVID-19 cases with different severity levels. We used beta distribution for the share of inpatients considered close contacts of HCWs and the reduction in the health workforce productivity, with a standard deviation assumed to be 20% of the mean. Beta distribution was also applied to the elasticities of MMR and U5MR, with standard deviations obtained from the literature.³⁶ We used gamma distribution for treatment costs with standard deviations assumed to be 20% of the means,⁴³ as their standard deviations were not available. The cost of meals and travel was not included in the stochastic sensitivity analysis given their small share in the total economic burden. The 2.5th percentile and 97.5th percentile were obtained after we ran 10,000 iterations for these parameters, as the 95% CIs.”*

The result section is updated accordingly.

Page 6; Data sources: It looks like some key data (e.g. treatment cost for mild/moderate cases) was extrapolated from one country to another. Please could you include a few sentences describing your methods for extrapolation.

Response: Thanks for the suggestion. As suggested, we have added the following in the main text on cost extrapolation: *“and f) parameter extrapolation from known sources, such as treatment costs for COVID-19 cases with various severity levels. Kenya was the only study site with an estimated treatment cost for home care for mild-moderate cases of COVID-19. Thus, the ratio of the treatment cost for home care to that for facility care among mid-moderate cases in Kenya was used to extrapolate the treatment cost of home care in other study sites. Similarly, if there were other unknown treatment costs in other sites, the corresponding cost ratios in Kenya were used to estimate the costs.”*

Table 2: It might be useful to see the total number of COVID-19 infections broken down into mild/moderate vs. severe cases, if this data are available.

Response: Thanks for the suggestion. In terms of the number of mid/moderate vs severe cases vs critical cases, we used a fixed ratio of 81%: 14%: 5% from the literature. Thus, we made the following note under Table 2: *“Note: The share of mild/moderate, severe, and critical COVID-19 cases was 81%, 14%, and 5%, respectively.”*

Table 3: Please add a label to note the values are in millions, and clarify currency year.

Response: as suggested, we have added them. The title of the table reads “*Table 1. Estimated economic burden by pathway in 2020 US\$ millions (percentage)*”

Table 4: I find it slightly misleading to present the low impact scenario and high impact scenario in parentheses in this table – as these are results from your scenario analyses rather than a representation of confidence intervals or certainty from your estimates. I would suggest to present your scenario analyses separately from your base case analysis, but also to provide a range of cost estimates e.g. using ranges in cost inputs to your model.

Response: As suggested, we have separated the scenario analysis from the base analysis and added 95% CIs for the base analysis.

Discussion: The analysis is very broad-brush, using a combination of primary and secondary data, supplemented with assumptions where no data exists. While the breadth of the analysis is not an issue in itself, it would be good to make this very clear in the presentation and discussion of results to avoid misinterpretation by policymakers and other stakeholders. I’d suggest revising some wording in the discussion to make clear that these results are not a presentation of scientific fact, but rather a modelling analysis designed to identify the potential scale of the issue and important considerations in responding to the pandemic.

Response: Thanks. In the title, we added “*A modelling study*” to make it explicit that the estimation is from economic modelling. We also follow the reviewer’s suggested and revised some wording in the discussion. For example, we started with “*our modelled results show that ...*” in the third paragraph of the discussion.

Discussion: I would suggest to make clear throughout the paper that this analysis was done using data from the first year of the pandemic. Many things have changed very rapidly since then (including infection control measures, infectiousness of COVID-19 strains, vaccine availability etc.) and the timeframe of the analysis is very important for interpretation of results. It would be useful to add a reflection as to what has changed in the last couple of years, and what the results of the analysis might mean in terms of policy today.

Response: Thanks for the suggestion. We have added in the first paragraph of the discussion section the phrase: “*in the first year of the pandemic*”.

We also follow the reviewer’s suggestions and added one paragraph to reflect the changes. The added paragraph reads: “*Please note that all estimates presented in this paper pertain to the first year of the pandemic when there were substantial shortages of PPE, COVID-19 vaccination coverage among both the general population and HCWs was extremely low, and the capacity of health systems in some countries to respond to COVID-19 was quite limited. All these factors contributed to the potentially high economic costs presented in this paper. Since then, many circumstances have improved. With reduced virulence of the virus, much greater vaccine*

coverage, and enhanced treatment and testing capacities, the economic burden of HCW infections in subsequent years is likely to be substantially lower.”

REVIEWERS' COMMENTS

Reviewer #1 (Remarks to the Author):

I am happy with the revisions undertaken by the authors in this resubmitted version of the paper. All of my comments have been engaged with and appropriate modifications carried out where possible, and, if not, suitable explanations provided. I recommend this revised paper for publication.

Reviewer #2 (Remarks to the Author):

The authors have satisfactorily replied to the comments raised on the previous version of the manuscript. The authors also conducted a stochastic analysis, which I think was needed to reflect the uncertainty of the parameters and allowed to present the results with a confidence interval.

I would only include, potentially in the discussion or, alternatively, I would make this clearer in the introduction, the fact that the model does not reflect within-country or within-site variation as average parameters/one parameter per site have been used for each site.

Reviewer #3 (Remarks to the Author):

Thank you for the edits that you have made to the paper. I am happy with all responses and have no further comments.

The authors have satisfactorily replied to the comments raised on the previous version of the manuscript. The authors also conducted a stochastic analysis, which I think was needed to reflect the uncertainty of the parameters and allowed to present the results with a confidence interval.

I would only include, potentially in the discussion or, alternatively, I would make this clearer in the introduction, the fact that the model does not reflect within-country or within-site variation as average parameters/one parameter per site have been used for each site.

Response: Thanks for the comment. As suggested, we have added the following sentences to the discussion: "While the model's estimates provide the total economic burden for each site, they don't account for variations within countries or individual sites. This is because the parameters used in the model represent site averages."